# An Asymmetric Latent Factorization-of-Tensors Model for Relation Analysis

Weiling Li [1]   Zhaoheng Shi [1]   Jiajia Mi [2]   Zhigang Liu [1]   Jialiang Wang [3]   Xin Luo [3]

## Abstract

Latent Factorization-of-Tensors (LFT) models are an effective approach for relation analysis. Existing LFT models assume each mode of the target tensor corresponds to an entity set and the relationships between entity sets are bipartite graphs to explore the relationships among entities within a mode. However, when the topological structure of entities in a mode is known, for example, entities are ordered physical quantities, such as time or coordinates, the relations between such modes forms a more complicated structure, i.e., aligned bipartite networks, and existing LFT models cannot accurately capture this structure. This work is the first to recognize and analyze this issue, and proposes an Asymmetric Latent Factorization-of-Tensors (ALFT) model to address it. ALFT can model aligned bipartite networks in mode pairs of a tensor by imposing constraints between particular mode pairs in the tensor network. Experimental results on real-world datasets demonstrate the existence of this issue and confirm that the proposed ALFT model can effectively resolve it.

## 1. Introduction

Tensorized relational data contain rich entity interaction information (Liu et al., 2025). However, feature extraction of associated entities from tensorized relational data remains a challenging task since it is commonly high-dimensional and incomplete (HDI) (Wu et al., 2025; Tran & Nguyen, 2024). Latent Factorization-of-Tensors (LFT) Models (Ahn et al., 2022; Xu et al., 2025a) have shown a promising ability to capture low-rank features of entities by approximat-

ing their HDI relations, making them widely used in tasks like building Recommendation Systems (Tiet et al., 2024), Knowledge Graph Completion (Yue et al., 2025), and Social Network Analysis (Fernandes et al., 2021).

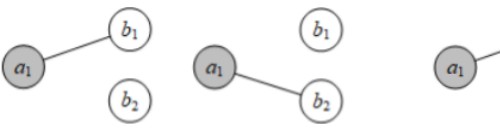 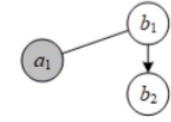

*(a)* Assuming entities are grouped by set, graph isomorphism exists.

*(b)* An interactive bipartite graph can be labeled.

*Figure 1.* Motivation of this work

To achieve more effective feature extraction, LFT models have been extensively extended, including employing more expressive latent feature (LF) spaces (Hosseinmardi et al., 2023) or designing task-specific constraints. All these efforts aim to preserve the intrinsic patterns within the data. For instance, a symmetric LF space can improve the prediction accuracy of protein-protein interactions, and the introduction of non-negativity constraints significantly boosts the performance of recommendation systems. Unfortunately, most LFT models assume that relation between any two modes is a bipartite graph and analyze the intrinsic structure of entity sets based on known connections to perform relation analysis. For instance, (Wu & Luo, 2021) proposes a CANDECOMP/PARAFAC (CP)-decomposition-based LFT model for missing Quality-of-Service data prediction and (Mi et al., 2023) utilizes a Tucker-decomposition-based LFT model for traffic flow completion.

Note that in real-world relations, only certain types of entity are organized in the form of sets, while others inherently possess specific structures, e.g., entities like time slots or temperature sequences. In fact, these modes are entity graphs. Thus, using bipartite graphs to describe the relational patterns between different mode types, i.e., entity set and entity graph, limits the performance of existing LFT models. It is easily explained with graph isomorphism theory. Assuming that relations between two modes is a bipartite graph, e.g., $G$, and there are two entity sets, e.g., $A$ and $B$, when $B$ is a set, elements in $A$ and $B$ form a non-labeled bipartite graph (Giamphy et al., 2023) as illustrated in Figure 1(a). When it is not surjective, graph isomorphism exists, e.g., $\{a_1, b_1\} = \{a_1, b_2\}$. When $B$ is not a set, as shown

[1]Department of Computer Science and Technology, Dongguan University of Technology, Dongguan, China [2]School of Computer Science and Technology, Guangdong University of Technology, Guangzhou, China [3]College of Computer and Information Science, Southwest University, Chongqing, China. Correspondence to: Zhigang Liu <liuzhigang@dgut.edu.cn>, Xin Luo <luoxin@swu.edu.cn>.

*Proceedings of the 43rd International Conference on Machine Learning*, Seoul, South Korea. PMLR 306, 2026. Copyright 2026 by the author(s).

in Figure 1(b), $G$'s structure is unique, which should be maintained or the extracted relation features may rely on erroneous topological assumptions. Obviously, it is believed that designing suitable LF spaces for interactions between different mode pairs, in accordance with their types, is an approach to enhance the representation learning performance of an LFT model.

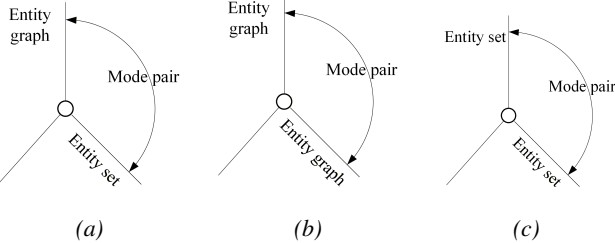

*Figure 2.* Examples of pair of modes in different scenarios

Moreover, for any high-order tensors, interactions between any pair of modes can occur in three scenarios, examples represented using a third-order tensor are shown in Figure 2. How to select suitable LF spaces and organize them appropriately remains an issue that needs to be addressed.

Aiming at addressing the above issues, this paper proposes an Asymmetric Latent Factorization-of-Tensors (ALFT) model, which is inspired by the Fully-connected tensor network (FCTN) (Zheng et al., 2021) and Singular Value Decomposition (SVD) (Cong & Zhou, 2023). For any two modes, if they are entity graphs, a learnable diagonal matrix between their LF spaces will be activated to maintain the relational topological structure between them.

This paper makes several unique contributions including

a) For the first time, it is proposed that there are different types of relational tensor mode, and a differentiated LF space is desired to represent different relation patterns;

b) An asymmetric tensor network (ATN) is proposed to adapt to different interaction patterns;

c) Extensive experiments on real-world datasets are conducted to evaluate the correctness of this finding and the performance of the proposed ALFT model.

The remainder of this paper is organized as follows. Section II states the preliminaries. Section III presents the ALFT model. The experimental results are discussed in Section IV. Finally, Section V concludes this paper.

## 2. Preliminaries

### 2.1. Latent Factorization-of-Tensors

Given an $N$-mode relation tensor $A \in \mathbb{R}^{|I_1| \times |I_2| \times \cdots \times |I_N|}$ that records known interactions $K$, where $|I_k|$ denotes the

number of entities in mode $k$. An LFT model minimizes the Euclidean distance between $A_i$ and its approximate, e.g., $\hat{A}_i$ on $K$, as follows:

$$S = \frac{1}{2} \sum_{A_i \in K} (A_i - \hat{A}_i)^2, \tag{1}$$

where $\hat{A}_i$ is obtained by reconstructing the results of the tensor factorization of $A_i$.

### 2.2. Tensor Network

Different tensor factorization methods shape distinct feature space structures, which ultimately determine the performance of LFT models. Mainstream tensor factorization methods, such as CP decomposition and Tucker decomposition, have demonstrated excellent performance in relation analysis. These models typically use a single core feature space to describe interactions among all modes. However, such an approach struggles to decouple interactions between different mode pairs. To address this limitation, researchers have subsequently proposed tensor factorization methods like tensor ring, tensor train, and FCTN. Among them, FCTN, which constructs a feature space for each mode in the form of a tensor of the same order as the original tensor, has shown promising performance and scalability in relation analysis (He et al., 2025; Han et al., 2024), FCTN approximate observed tensor entries as follows:

$$\hat{\mathbf{A}}(i_1, i_2, \ldots, i_N) =$$
$$\sum_{l_{1,2}=1}^{L_{1,2}} \sum_{l_{1,3}=1}^{L_{1,3}} \cdots \sum_{l_{1,N}=1}^{L_{1,N}} \sum_{l_{2,3}=1}^{L_{2,3}} \cdots \sum_{l_{2,N}=1}^{L_{2,N}} \cdots \sum_{l_{N-1,N}=1}^{L_{N-1,N}}$$
$$T_1(i_1, l_{1,2}, l_{1,3}, \ldots, l_{1,N}) T_2(l_{1,2}, i_2, l_{2,3}, \ldots, l_{2,N})$$
$$\cdots T_k(l_{1,k}, l_{2,k}, \ldots, l_{k-1,k}, i_k, l_{k,k+1}, \ldots, l_{k,N})$$
$$\cdots T_N(l_{1,N}, l_{2,N}, \ldots, l_{N-1,N}, i_N), \tag{2}$$

where $T_i(\cdot)$ is a specified latent factor in the $i$-th LF tensor.

## 3. Methodology

### 3.1. Asymmetric Tensor Network

Recall that existing LFT models assume that the entities corresponding to each mode of a relation tensor are grouped into sets, while also presuming that the interaction between any two modes constitutes a bipartite graph, e.g., $G_B = \{U, I, E\}$, to optimally explore potential interaction patterns by enumerating all graph isomorphisms. However, when entities in multiple modes are not grouped as simple sets, i.e., when entities of a mode exhibit explicit internal structures, the interaction between these modes essentially forms an aligned bipartite network.

**Definition 1: aligned bipartite network.** Assume there are two homogeneous networks, e.g., $G_U = \{U, E_U\}$ and

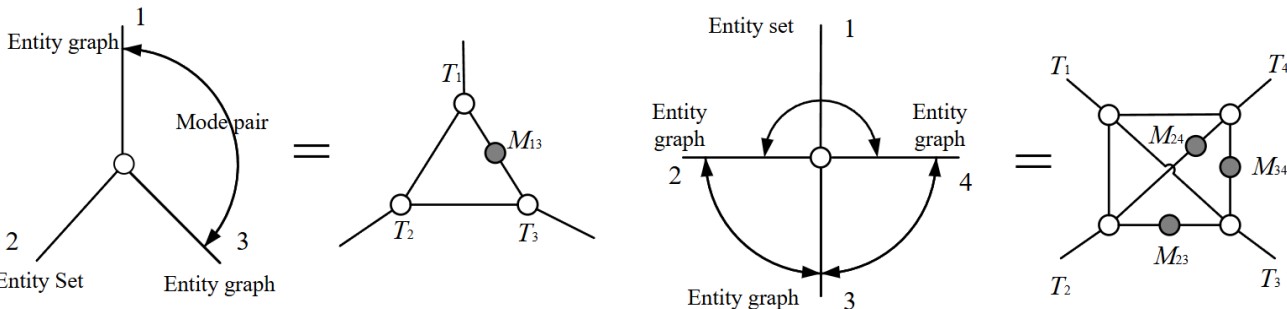

*Figure 3.* Example of Asymmetric Tensor Network for third-order and fourth-order interaction tensors

$G_I = \{I, E_I\}$, and a bipartite graph $G_B = \{U, I, E\}$ exists, $G = \{G_B, G_U, G_I\}$ is an aligned bipartite network.

It is evident that aligned bipartite networks are subject to stricter constraints. Preserving internal structures, i.e., $G_U$ and $G_I$, thus helps avoid simplistic topological isomorphism and serves as the key to enhancing the representational capacity of LFT models. Therefore, while three distinct types of mode pairs may exist as shown in Figure 2, we mainly need to consider the case where the modes are entity graphs, i.e., which possesses explicit internal structures.

However, how to achieve differentiated representation for distinct mode pairs during the tensor factorization process remains a critical challenge to be addressed in this work. To address this issue, we may consider reverting to a simpler scenario, i.e., matrix factorization. In the context of matrix factorization, SVD and Latent Factor (LF) (Tegene et al., 2023; Aslanyan & Frasincar, 2021; Merabet & Benmerzoug, 2022) models are two commonly used methods. Among them, the LF model is similar to the LFT model, it learns the features of entities within each set by analyzing known interactions. Compared to the LF model, the features of SVD possess clear global geometric significance, enabling it to effectively capture the global structure. Mathematically, the existence of the diagonal matrix in SVD ensures that the low-rank approximation is optimal in the sense of the Frobenius norm. However, since SVD is an orthogonal unitary equivalence decomposition that requires the row and column orders of the orthogonal matrices to be preserved, it cannot be directly applied in tensor networks.

Consider the low-rank decomposition of $R \in \mathbb{R}^{m \times n}$ without orthogonality constraints, i.e., $R \approx AC$. There exist infinitely many equivalent decomposition forms $(A, C) \sim (AD, HC)$ with $DH = \mathbf{I}$, where $A \in \mathbb{R}^{m \times f}, C \in \mathbb{R}^{f \times n}, D \in \mathbb{R}^{f \times f}, H \in \mathbb{R}^{f \times f}$ are matrices and $\mathbf{I}$ is an identity matrix, thus making the identities of the rows impossible to be uniquely determined. Rows and columns are not fixed but drift with $D$ and $H$, corresponding to gauge degrees of freedom with $f^2$ parameters. Let $B$ be an invertible diagonal matrix with distinct diagonal entries, Without orthogonality constraints, the gauge degrees of freedom of

the factorization $R = ABC$ are reduced from the general linear group $GL(f)$ with $f^2$ parameters to the generalized permutation transformations, whose continuous degrees of freedom have dimension $2f$. This provides a theoretical foundation and inspires us to propose the Asymmetric Tensor Network (ATN). Examples of ATN for third- and fourth-order relational tensors are illustrated in Figure 3.

Specifically, the ATN inserts learnable diagonal matrices, e.g., $M_{p,q}$, between the latent feature tensors corresponding to modes $p$ and $q$, preserving the interaction topology between the corresponding mode pairs through an SVD-like structure(Zheng et al., 2024) and $M_{p,q}$ is activated exclusively when both of its linked modes are categorized as the entity graph type. The indices $p$ and $q$ range over all pairs of distinct modes, i.e., $1 \le p < q \le N$, ensuring that each ordered edge between modes is modeled with a dedicated diagonal matrix. An approximation can be obtained with the ATN as follows:

$$\hat{\mathbf{A}}(i_1, i_2, \ldots, i_N) =$$

$$\sum_{l_{1,2}=1}^{L_{1,2}} \sum_{l_{1,3}=1}^{L_{1,3}} \cdots \sum_{l_{1,N}=1}^{L_{1,N}} \sum_{l_{2,3}=1}^{L_{2,3}} \cdots \sum_{l_{2,N}=1}^{L_{2,N}} \cdots \sum_{l_{N-1,N}=1}^{L_{N-1,N}}$$

$$M_{1,2}(l_{1,2}, l_{1,2}) \, M_{1,3}(l_{1,3}, l_{1,3}) \cdots M_{1,N}(l_{1,N}, l_{1,N})$$

$$M_{2,3}(l_{2,3}, l_{2,3}) \cdots M_{2,N}(l_{2,N}, l_{2,N}) \cdots$$

$$M_{N-1,N}(l_{N-1,N}, l_{N-1,N})$$

$$\big( T_1(i_1, l_{1,2}, l_{1,3}, \ldots, l_{1,N}) \, T_2(l_{1,2}, i_2, l_{2,3}, \ldots, l_{2,N}) \cdots$$

$$T_k(l_{1,k}, l_{2,k}, \cdots, l_{k-1,k}, i_k, l_{k,k+1}, \ldots, l_{k,N}) \cdots$$

$$T_N(l_{1,N}, l_{2,N}, \ldots, l_{N-1,N}, i_N) \big),$$

$$(3)$$

where $T_k \in \mathbb{R}^{L_{1,k} \times L_{2,k} \times \cdots \times L_{k-1,k} \times I_k \times L_{k,k+1} \times \cdots \times L_{k,N}}$ represents latent feature tensor for $k = 1, 2, \ldots, N$, and $M_{p,q} \in \mathbb{R}^{L_{p,q} \times L_{p,q}}$ is the diagonal factor matrix for all $\{p, q\} \in \{1, 2, 3, \ldots, N\}$. In particular, $M_{p,q}$ is set as an identity matrix when one of its associated modes is an entity set.

## 3.2. Objective Function

This work employs Euclidean distance to measure the difference between the data reconstructed by ATN and the known data and L2 regularization to avoid overfitting. It is worth noting that, unlike common symmetric tensor factorization models, ATN may not satisfy the property where local optima coincide with global optima due to its asymmetric structure (Jin et al., 2017). Therefore, its performance is more sensitive to data distribution. To avoid overfitting while accounting for entity frequency imbalance, we adopt a sample-aware weighted L2 regularization strategy.

For tensor $T_k$ , the regularization weight associated with entity $i_k$ is defined as

$$\omega_{k,i} = \frac{|I_{i_k}|}{\max_{j \in \mathcal{E}_k} |I_j|}, \tag{4}$$

where $|I_{i_k}|$ denotes the number of observed samples associated with entity $i_k$, and $\mathcal{E}_k$ denotes the entity set of mode $k$.

The regularization term of $M_{p,q}$ is set as $\bar{\omega}$, the average of the per-mode mean weights across all modes, to avoid penalty fluctuations. Formally:

$$\bar{\omega} = \frac{1}{N} \sum_{k=1}^{N} \bar{\omega}_k, \quad \text{where} \quad \bar{\omega}_k = \frac{1}{|\mathcal{E}_k|} \sum_{i \in \mathcal{E}_k} \omega_{k,i}. \tag{5}$$

By combining ATN with differentiated imbalanced regularization terms, the objective function of ALFT in the element-wise form is as follows:

$$\begin{aligned}
\mathcal{S} =& \frac{1}{2} \sum_{(i_1,\ldots,i_N) \in K} \Big( A(i_1,\ldots,i_N) - \hat{A}(i_1,\ldots,i_N) \Big)^2 \\
& + \frac{\lambda}{2} \sum_{k=1}^{N} \sum_{i_k=1}^{I_k} \omega_{k,i_k} \|T_k(i_k)\|_F^2 \\
& + \frac{\lambda}{2} \bar{\omega} \sum_{1 \le p < q \le N} \|M_{p,q}\|_F^2 ,
\end{aligned} \tag{6}$$

where $K$ is the known element entries set, $T_k(i_k)$ denotes the latent subtensor obtained by fixing the entity index $i_k$ in tensor $T_k$, and $\lambda$ is a tunable regularization coefficient.

## 3.3. Optimization

It is evident that solving ALFT constitutes a large-scale non-convex optimization problem, numerous scholars have conducted extensive research on it. Given that the primary objective of this paper is to reveal how the performance of existing LFT models is affected by erroneous graph isomorphism and to address this issue from the perspective of constructing a latent tensor space, employing the simplest

mini-batch gradient descent method to solve ALFT is sufficient for validating the effectiveness of ALFT. For a known relation $a = A(i_1,\ldots,i_N)$, its approximation $\hat{a}$ computed by ATN can be abbreviated as

$$\hat{a} = \sum_{\{l_{p,q}\}} \Big( \prod_{1 \le p < q \le N} M_{p,q}(l_{p,q},l_{p,q}) \Big) \Big( \prod_{k=1}^{N} T_k(\mathbf{l}_k, i_k) \Big), \tag{7}$$

where $\mathbf{l}_k = (l_{1,k},\ldots,l_{k-1,k},l_{k,k+1},\ldots,l_{k,N})$. Then the gradient with respect to the latent tensor element $T_k(\mathbf{l}_k, i_k)$ can be obtained as follows:

$$\begin{aligned}
\frac{\partial \mathcal{S}}{\partial T_k(\mathbf{l}_k, i_k)} =& \\
(\hat{a} - a) &\sum_{\{l_{p,q}\}/\mathbf{l}_k} \Big( \prod_{1 \le p < q \le N} M_{p,q}(l_{p,q},l_{p,q}) \Big) \\
& \times \Big( \prod_{m \ne k} T_m(\mathbf{l}_m, i_m) \Big) + \lambda \omega_{k,i_k} T_k(\mathbf{l}_k, i_k),
\end{aligned} \tag{8}$$

where $\{l_{p,q}\}/\mathbf{l}_k$ denotes all latent indices expect $\mathbf{l}_k$. Similarly, the gradient with respect to the diagonal interaction parameter $M_{p,q}(l_{p,q},l_{p,q})$ can be obtained as follows:

$$\begin{aligned}
\frac{\partial \mathcal{S}}{\partial M_{p,q}(l_{p,q},l_{p,q})} =& \\
(\hat{a} - a) &\sum_{\{l_{u,v}\}/l_{p,q}} \Big( \prod_{(u,v) \ne (p,q)} M_{u,v}(l_{u,v},l_{u,v}) \Big) \\
& \times \Big( \prod_{k=1}^{N} T_k(\mathbf{l}_k, i_k) \Big) + \lambda \bar{\omega} M_{p,q}(l_{p,q},l_{p,q}).
\end{aligned} \tag{9}$$

## 4. Experiments

To comprehend ALFT's performance, three groups of experiments are conducted to answer the following questions:

Q1: Is ATN capable of accommodating different relation patterns?

Q2: What is the applicability of ALFT?

Q3: How does ALFT perform compared to state-of-the-art models?

### 4.1. Experiment Settings

**Datasets.** Table 1 presents the detailed information of the datasets used in the experiments. To evaluate the model's stability and generalization, we divide the $n$-th dataset, e.g., D$n$, where $n = 1, 2, 3$, into four split settings, denoted as D$n$.1, D$n$.2, D$n$.3, and D$n$.4, corresponding to train:validation:test ratios as 50%:20%:30%, 60%:20%:20%, 70%:10%:20%, and 80%:10%:10%, respec-

tively.

Table 1. Involved Datasets

| Dataset | Shanghai (D1) (Mi et al., 2023) | Guangzhou (D2) (Chen et al., 2018) | Aarhus (D3) (Baggag et al., 2019) |
|---|---|---|---|
| Road Segments (mode 1) | 18 | 214 | 449 |
| Date (mode 2) | 28 | 61 | 117 |
| Time Segments (mode 3) | 288 | 144 | 288 |

To provide a more comprehensive answer to Q1, in addition to the datasets mentioned above, we also utilized other datasets, including UMLS (D4) and Kinship (D5) (Kemp et al., 2006), MovieLens100k (D6) (Harper & Konstan, 2015),and a sparse video dataset collected by ourselves (D7).

**Evaluation Metrics.** In this work, the reconstruction accuracy of relational tensors is utilized to indirectly measure the relation analysis performance of involved models, and Root Mean Squared Error (RMSE) and Mean Absolute Error (MAE) are adopted.

**Implement details.** Embedding dimension $d$ is set to 20. Patience value is set to 20 epochs for early stopping, i.e., training stops if MAE on the validation set does not improve for 20 consecutive epochs. Owing to differences in model architectures and data splitting scenarios, the hyperparameter configurations vary accordingly. Python 3.9.23, PyTorch 2.8.0 and CUDA 12.8 are utilized to implement the experiments. All experiments are conducted on a server with Intel(R) Core(TM) Ultra 7 255HX processor, 32GB memory, NVIDIA GeForce RTX 5070 Ti.

**Involved Models.** To evaluate the performance of our proposed model, state-of-the-art baseline models are selected for comprehensive comparison. Table 2 briefly summarizes all the involved models.

### 4.2. Experiment Group One

The objective of experiment group one is to address Q1. It comprises two experiments. **Experiment 1.1** aims to verify the presence of aligned bipartite networks in relational tensors and examine the effectiveness of the proposed ATN model by employing different tensor factorization models. **Experiment 1.2** serves as a supplementary study. Given that date mode and time fragment mode carry temporal information and could theoretically be integrated, this additional experiment aims to demonstrate whether preserving the independence of the two modes contributes to better relation analysis.

**Quantitative results of Experiment 1.1.** In this experiment, ALFT, FCTN, SVDinsTN, and three tensor network

Table 2. Involved Models

| Model | Description |
|---|---|
| FCTN (Zheng et al., 2021) | A fully-connected tensor network decomposition model. |
| SVDinsTN (Zheng et al., 2024) | An SVD-inspired tensor network decomposition model. |
| NCF (He et al., 2017) | A neural network-based collaborative filtering framework. |
| AEMC-NE (Fan et al., 2024) | A neuron-enhanced autoencoder matrix completion model. |
| AFDGCF (Wu et al., 2024) | An adaptive feature decorrelation graph collaborative filtering model. |
| GLCPN (Bi et al., 2025) | A graph linear convolution pooling network. |
| TCA (Su et al., 2021) | A tensor completion model with a multidimensional tensor via a reference reporting mechanism. |
| BCGP (Chen et al., 2019) | A Bayesian probabilistic tensor factorization model based on Bayesian inference. |
| TUCKER (Mi et al., 2023) | A latent tensor factorization model based on Tucker decomposition. |
| STD (Chen et al., 2018) | A tensor decomposition model combined with singular value decomposition (SVD). |
| DAIN (Oh et al., 2021) | A data augmentation model with an influence mechanism for tensor learning. |
| FreTS (Yi et al., 2023) | A time time series forecasting model based on frequency-domain MLP. |
| ALFT | Our model. |

models with specially designed constraint placements are employed to represent the target tensor. These models collectively cover all potential types of constraint configurations. The RMSE and MAE achieved by involved models on D2 are listed in Table 3. It is evident that ALFT consistently achieves state-of-the-art performance across all evaluated datasets and metrics. For example, on D2.1, the RMSE of ALFT represents a substantial improvement of up to 5.3% over FCTN's 3.613 and 8.4% over SVDinsTN's 3.737.

Datasets D4-D7 impose different requirements on the constraints. Note that the primary purpose of Experiment 1.1 is to validate the effectiveness of ATN. Therefore, for datasets D4 and D5, we used cross-entropy to construct the objective function, rather than ALFT. For consistency, their results are evaluated using RMSE and MAE. The results are shown in Tables 4 and 5 . It can be observed from the results that the diagonal constraint does possess the ability to regularize the entity graph structure. On D7, where all modes correspond to entity graphs, diagonal matrices strengthen the model's structural constraint ability, so activating all diagonal matrices yields the best performance. In contrast, on D4, D5 and D6, with no or only one entity graph mode, diagonal matrices have a negative impact. These results complement

*Table 3.* Performance of different LF structures

| Model | FCTN | SVDinsTN | ALFT | Model-1 | Model-2 | Model-3 |
|---|---|---|---|---|---|---|
| Structure |  |  |  |  |  |  |

| | | FCTN | SVDinsTN | ALFT | Model-1 | Model-2 | Model-3 |
|---|---|---|---|---|---|---|---|
| RMSE | D2.1 | 3.613±0.014 | 3.737±0.035 | **3.423±0.019** | 3.550±0.030 | 3.470±0.010 | 3.697±0.042 |
| | D2.2 | 3.525±0.047 | 3.732±0.019 | **3.396±0.024** | 3.547±0.042 | 3.448±0.004 | 3.674±0.047 |
| | D2.3 | 3.390±0.018 | 3.713±0.037 | **3.349±0.024** | 3.461±0.033 | 3.391±0.020 | 3.662±0.048 |
| MAE | D2.1 | 2.415±0.006 | 2.471±0.018 | **2.305±0.009** | 2.368±0.015 | 2.334±0.005 | 2.433±0.020 |
| | D2.2 | 2.388±0.008 | 2.461±0.009 | **2.285±0.013** | 2.362±0.023 | 2.318±0.004 | 2.425±0.025 |
| | D2.3 | 2.340±0.003 | 2.453±0.017 | **2.262±0.009** | 2.319±0.013 | 2.288±0.008 | 2.420±0.027 |

*Table 4.* RMSE under different Activation Strategy

| Diagonal Matrix Activation Status | D4 | D5 | D6 | D7 |
|---|---|---|---|---|
| All diagonal matrices deactivated | **0.248** | **0.291** | **0.970** | 26.576 |
| One diagonal matrix activated | 0.521 | 0.553 | 0.977 | 26.510 |
| All diagonal matrices activated | 0.365 | 0.379 | 0.974 | **25.555** |

*Table 5.* MAE Under different Activation Strategy

| Diagonal Matrix Activation Status | D4 | D5 | D6 | D7 |
|---|---|---|---|---|
| All diagonal matrices deactivated | **0.168** | **0.228** | **0.758** | 18.728 |
| One diagonal matrix activated | 0.432 | 0.482 | 0.765 | 18.321 |
| All diagonal matrices activated | 0.278 | 0.335 | 0.762 | **17.849** |

our findings on the transportation dataset, where two modes correspond to entity graphs, and activating one diagonal matrix achieved the best performance.

**Quantitative results of Experiment 1.2.** In this experiment, the original third-order tensor is converted into a matrix by merging the two time-correlated modes into a unified temporal dimension for training matrix-oriented relation analysis models, namely NCF, AEMC-NE, AFDGCF and GLCPN. The experimental results show that ALFT outperforms all matrix-oriented relation analysis models on all metrics. Detailed experimental results are presented in Figure 4. For example, ALFT achieves the lowest RMSE at 5.54, improving upon NCF's 6.12 by 9.5%, GLCPN's 6.18 by 10.4%, AFDGCF's 6.50 by 14.8%, and AEMC-NE's 8.49 by 34.8%. Experimental results indicate that for two modes of the same type, although they can be merged into a single mode, the merged data, which lacks one of the original modes, struggles to express the rich internal interactive logic. This information degradation directly leads to a notable decline in model performance. Therefore, in similar scenarios, using ALFT to preserve the aligned bipartite net-

work structure yields better outcomes than compressing the modes.

**Remark 1. The presence of an aligned bipartite network structure in relational tensors can enhance the accuracy of relation analysis when constraints are applied to the relevant structure during the analysis process.** During the tensor factorization process with ATN, constraints are imposed between the entity features corresponding to the respective modes to understand involved aligned bipartite networks. The results of Experiment 1.1 demonstrate the validity of our finding.

From the experimental results, it can be observed that neither FCTN, which treats all mode pairs as bipartite graphs, nor models employing other constraint configuration strategies can match the performance of ALFT. This is because, among the three modes in the experimental data, the two time-related modes are entity graphs and the mode of road segments is an entity set. Therefore, the relation between the time-related modes is an aligned bipartite network, whose structure should be maintain during the learning process to maintain a stable relationship between them. In other words, adopting ATN's constraint strategy effectively preserves the inherent structural stability of the data. Notably, compared to FCTN, Model-1 and Model-2 also achieve relatively high metric scores. This result is explainable since the relationship between the two modes is neither purely a bipartite graph nor strictly an aligned bipartite network, the effect of imposing constraints on relation analysis may be either positive or negative. However, as erroneous constraints increase, the interaction between modes becomes confined to incorrect structures, leading to a significant decline in performance, for example, SVDinsTN and Model-3 perform bad in this experiment.

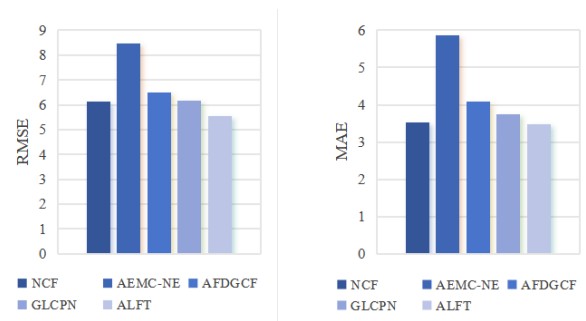

Figure 4. Results of Experiment 1.2

## 4.3. Experiment Group Two

The objective of experiment group two is to address Q2. It comprises two experiments. **Experiment 2.1** investigates how changes in the number of entities contained in the modes affect ALFT's performance, while **Experiment 2.2** aims to explore the impact of data density on ALFT. Notably, the trends in model performance under these two data attribute variations can also indirectly corroborate the issue raised in Q1 regarding whether erroneous graph isomorphism affects LFT.

**Quantitative results of Experiment 2.1.** In this experiment, we vary the number of entities in the entity set, i.e., mode 1, and the entity graph, i.e., mode 2, separately. The results are visualized in Figures 5 and 6, respectively. When reducing the entity number in mode 2, we observe that the performance gap between ALFT and FCTN narrows progressively, and FCTN eventually overtakes ALFT. For example, when 50% of mode 2 entities are removed, ALFT achieves an RMSE of 5.57, outperforming FCTN's 5.81 by 4.1%. However, when the removal ratio reaches 75%, FCTN overtakes ALFT with an RMSE of 6.04, outperforming ALFT's 6.15 by 1.8%. When reducing the entity number in mode 1, the performance gap between ALFT and FCTN widens continuously, with ALFT maintaining an increasingly significant lead. For instance, when the remaining mode 1 entity ratios are 79.17% and 50.15%, ALFT's MAE outperforms FCTN by 3.8% and 5%, respectively.

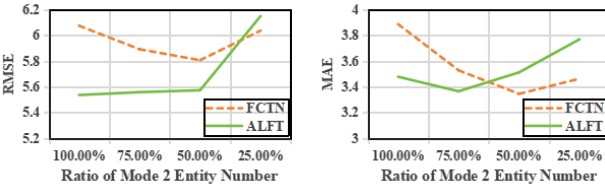

Figure 5. RMSE and MAE with varying Entity number ratios of Mode 2

**Quantitative results of Experiment 2.2.** In this experiment, we set different data densities to assess changes in the performance of all models across varying data densities. It is observed that as the data density increases, the

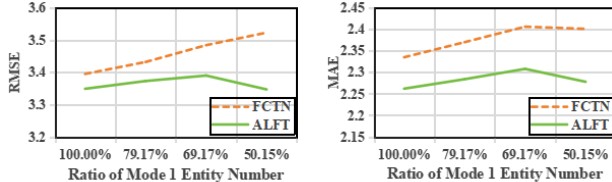

Figure 6. RMSE and MAE with varying Entity number ratios of Mode 1

performance of all involved models gradually improves, and ALFT consistently outperforms all other compared models across all evaluation metrics. Results under varying data densities align with this trend and the details are in Tables 6 and 7. For example, for dataset D1.4, the experimental result is presented as follows: ALFT achieves an RMSE of 5.536, representing a substantial improvement of 8.9% over FCTN's 6.075 and 4.6% over SVDinsTN's 5.803.

Table 6. RMSE achieved in different Data Densities

| Dataset | ALFT | FCTN | SVDinsTN |
|---------|------|------|----------|
| D1.1 | **6.592±0.054** | 7.432±0.023 | 6.623±0.092 |
| D1.2 | **6.188±0.065** | 6.820±0.091 | 6.191±0.059 |
| D1.3 | **5.854±0.057** | 6.374±0.060 | 5.922±0.155 |
| D1.4 | **5.536±0.148** | 6.075±0.161 | 5.803±0.089 |
| D2.1 | **3.499±0.007** | 3.673±0.010 | 3.745±0.025 |
| D2.2 | **3.423±0.019** | 3.613±0.014 | 3.737±0.035 |
| D2.3 | **3.396±0.024** | 3.525±0.047 | 3.732±0.019 |
| D2.4 | **3.349±0.024** | 3.390±0.018 | 3.713±0.037 |

Table 7. MAE achieved in different Data Densities

| Dataset | ALFT | FCTN | SVDinsTN |
|---------|------|------|----------|
| D1.1 | **4.350±0.054** | 5.020±0.017 | 4.394±0.064 |
| D1.2 | **3.984±0.041** | 4.503±0.072 | 4.029±0.057 |
| D1.3 | **3.721±0.027** | 4.165±0.031 | 3.858±0.159 |
| D1.4 | **3.478±0.048** | 3.887±0.072 | 3.767±0.047 |
| D2.1 | **2.343±0.004** | 2.442±0.003 | 2.473±0.011 |
| D2.2 | **2.305±0.009** | 2.415±0.006 | 2.471±0.018 |
| D2.3 | **2.285±0.013** | 2.388±0.008 | 2.461±0.009 |
| D2.4 | **2.262±0.009** | 2.340±0.003 | 2.453±0.017 |

**Remark 2. The performance of ALFT is positively correlated with the ratio of the number of entities in the entity graph to that in the entity set.** It is worth noting that a bipartite graph and an aligned bipartite network may share a mode. Consequently, the constraints of the aligned bipartite network can affect the bipartite graph. Therefore, not all constraints imposed on aligned bipartite networks yield a positive effect, and a trade-off exists when using ALFT.

Experimental results indicate that the performance of ALFT is positively correlated with a ratio coefficient, e.g., $\gamma$, which

*Table 8.* RMSE achieved by involved models

| Dataset | ALFT | FCTN | SVDinsTN | TUCKER | BCGP | FreTS | STD | TCA | DAIN |
|---|---|---|---|---|---|---|---|---|---|
| D1.1 | **6.188±0.065** | 6.820±0.091 | 6.191±0.059 | 7.095±0.079 | 6.975±0.051 | 16.037±1.32 | 7.901±0.013 | 7.708±0.040 | 6.199±0.139 |
| D2.1 | **3.423±0.019** | 3.613±0.014 | 3.737±0.035 | 4.085±0.018 | 4.003±0.005 | 9.583±1.766 | 3.816±0.003 | 4.309±0.007 | 3.965±0.013 |
| D3.1 | **2.821±0.015** | 2.836±0.004 | 2.982±0.033 | 4.238±0.030 | 3.687±0.015 | 7.388±4.404 | 3.264±0.002 | 8.648±0.017 | 3.286±0.029 |
| D1.2 | **5.536±0.148** | 6.075±0.161 | 5.803±0.089 | 6.980±0.159 | 6.801±0.040 | 15.819±0.914 | 7.567±0.009 | 7.432±0.114 | 5.856±0.028 |
| D2.2 | **3.349±0.024** | 3.390±0.018 | 3.713±0.037 | 4.069±0.021 | 3.989±0.003 | 10.903±0.230 | 3.664±0.002 | 4.259±0.021 | 3.944±0.013 |
| D3.2 | **2.731±0.013** | 2.733±0.012 | 2.983±0.045 | 4.203±0.005 | 3.680±0.010 | 10.432±0.596 | 3.256±0.002 | 8.585±0.014 | 3.214±0.017 |

*Table 9.* MAE achieved by involved models

| Dataset | ALFT | FCTN | SVDinsTN | TUCKER | BCGP | FreTS | STD | TCA | DAIN |
|---|---|---|---|---|---|---|---|---|---|
| D1.1 | 3.984±0.041 | 4.503±0.072 | 4.029±0.057 | 4.860±0.060 | 4.923±0.039 | 13.376±1.322 | 5.610±0.008 | 5.312±0.043 | **3.930±0.128** |
| D2.1 | **2.323±0.007** | 2.415±0.006 | 2.471±0.018 | 2.714±0.005 | 2.640±0.006 | 8.544±1.879 | 2.543±0.002 | 2.866±0.003 | 2.613±0.004 |
| D3.1 | **1.529±0.012** | 1.542±0.002 | 1.585±0.015 | 2.131±0.021 | 1.831±0.006 | 6.001±5.102 | 1.742±0.001 | 3.199±0.007 | 1.726±0.079 |
| D1.2 | **3.478±0.048** | 3.887±0.072 | 3.767±0.047 | 4.774±0.065 | 4.773±0.028 | 13.734±0.807 | 5.391±0.010 | 5.143±0.045 | 3.653±0.098 |
| D2.2 | **2.262±0.009** | 2.340±0.003 | 2.453±0.017 | 2.702±0.007 | 2.628±0.003 | 10.064±0.253 | 2.448±0.002 | 2.822±0.009 | 2.610±0.029 |
| D3.2 | **1.501±0.009** | 1.507±0.004 | 1.584±0.016 | 2.109±0.005 | 1.832±0.008 | 9.343±0.626 | 1.740±0.001 | 3.100±0.005 | 1.668±0.010 |

is the ratio of the number of entities in the entity graphs to the number of entities in the entity sets. $\gamma$ can be estimated as follows:

$$\gamma = \frac{\prod\limits_{m \in \mathcal{M}} |\mathcal{V}_m|}{\prod\limits_{n \in \mathcal{N}} |\mathcal{V}_n|}, \qquad (10)$$

where $\mathcal{M}$ and $\mathcal{N}$ represent the indices of the modes corresponding to the entity graph and entity set, respectively, while $\mathcal{V}_m$ and $\mathcal{V}_n$ denote the sets of entities corresponding to their respective modes.

In our experiments, increasing the entity number in an entity graph enlarges $\gamma$, and thus yields a stronger constraint effect. Conversely, reducing the entity number in an entity graph diminishes $\gamma$, which weakens the topological structure fixation. This exacerbates the adverse impacts of graph isomorphism, as the model not only fails to learn the true temporal continuity across modes but also infers erroneous temporal information due to spurious mode permutations. Such a weak structural constraint effect ultimately degrades the overall model performance.

**Remark 3. The reconstruction performance of ALFT is positively correlated with data density, which is similar to the case in matrix factorization.**

The underlying reason is that its SVD-inspired structure can capture the effective structural information of the target data, thereby enhancing the reliability of low-rank approximations. This result indicates that when learning higher-order

relational tensors, embedding constraints between mode pairs whose relationships can be regarded as aligned bipartite networks, i.e., forming an SVD-like structure, provides greater mathematical rigor in capturing the overall topological structure. Conversely, employing an unconstrained design for relationships involving associated entity sets is more effective in learning latent semantic features from interaction data.

### 4.4. Experiment Group Three

This experiment is primarily designed to answer Q3, i.e., demonstrating the superiority of ALFT. Due to the flexible tensor network structure, ALFT can improve the optimization accuracy in a certain degree compared with the other state-of-the-art model as shown in Table 8 and Table 9. For example, on D3.2, the MAE of ALFT is 1.501 with an accurate improvement of 0.4%, 5.2%, 28.8%, 18.1%, 83.9%, 13.73%, 51.6%, and 10.0% over FCTN-DAIN, respectively. While the RMSE of ALFT is 2.731, compared to 2.733 for FCTN, 2.983 for SVDinsTN, 4.203 for TUCKER, 3.680 for BCGP, 10.432 for FreTS, 3.256 for STD, 8.585 for TCA, and 3.21 for DAIN, the accuracy improves by 0.1%, 8.4%, 35.0%, 25.7%, 73.8%, 16.1%, 68.2%, and 15.0%, respectively. Although DAIN achieves a slightly lower MAE than ALFT on D1.1, this marginal advantage does not generalize to D2 and D3. Additionally, the time series model FreTS shows limited performance when applied to HDI data and struggle to model the complex spatio-temporal dependencies inherent in traffic flow datasets.

**Remark 4. The computational overhead of ALFT, in terms of both time and space, lies between that of FCTN**

**and SVDinsTN.** Due to the asymmetric structure of ATN, it exhibits sensitivity to hyperparameters. Therefore, in future work, we will employ more suitable optimization methods and parameter adaptation strategies to further enhance the performance of ALFT.

## 5. Limitations

While the proposed method, compared to existing baseline models, can effectively leverage entity-type distinctions in relational data for improved relational analysis, it is important to acknowledge that it still has certain potential limitations as follows:

a) The architecture of ATN restricts the application of orthogonal constraints; consequently, the model cannot yet fully constrain the mode-pair structures. In future work, we will explore more suitable tensor decomposition methods, e.g., tensor-ring decomposition (Chen et al., 2025), to enhance the model's capability for local structural constraints, and b) Although it is clear whether a mode corresponds to an entity set or an entity graph, it currently relies on manual judgment to adjust the embedding structure of ALFT.

## 6. Conclusion

This work is the first to identify that the performance of existing LFT models can be constrained by erroneous graph isomorphism assumptions, since the relationship between any two modes in a relational tensor may not only be a bipartite graph but also an aligned bipartite network. To demonstrate the existence of this problem and address it, this study proposes an asymmetric tensor network that stabilizes the interaction topology by incorporating learnable constraint matrices between mode pairs explicitly defined as aligned bipartite networks. Building on this, we introduce the Asymmetric Tensor Latent Feature Model. Empirical results on real-world datasets demonstrate the existence of this issue and show that the proposed method effectively addresses it.

### Acknowledgements

The authors sincerely thank the authors of FCTN and SVDinsTN. These models enable us to verify and resolve some limitations of traditional LFT models. This work was supported in part by NSFC under Grants W2412112 and 62102086, and in part by Guangdong Basic and Applied Basic Research Foundation under Grant 2024A1515140137.

### Impact Statement

This paper presents work whose goal is to advance the field of Machine Learning, and more specifically, the theoretical understanding of latent factorization-of-tensors as a tool for relation analysis tasks. There are many potential societal consequences of our work, none of which we feel must be specifically highlighted here.

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
