# OpenReview forum: "An Asymmetric Latent Factorization-of-Tensors Model for Relation Analysis"
_ICML.cc/2026/Conference — ICML 2026 regular_

### Official Review · Reviewer_kN7Q · 2026-03-03

[review text omitted: it was posted to a different submission]

---

> ### Author Rebuttal · Authors · 2026-03-30
>
> Thank you for your comment.
>
> 1.  To verify the correctness of the theory proposed in this paper, we have made the following efforts.
>
> Proposition 1. In a relational tensor, the entities corresponding to a mode may be either set or labeled entities. In R, the former manifests as rows and columns that can be permuted arbitrarily, whereas the latter have fixed rows and columns.
>
> Proposition 1 is self-evident, for example shuffling the temporal order of time-varying data or altering the rows and columns of an image corrupts the original data.
>
> Proving the theoretical correctness of the proposed method is equivalent to demonstrating that introducing a learnable diagonal matrix can enhance the model's ability to learn and preserve the structural invariance of the mode pair R. To simplify the proof, we assume that the parameter tensor is the contraction of a series of matrices, among which one matrix can characterize the relationship between the current mode and the other mode associated with R, and the relationship R can be approximated by a series of matrix multiplications.
>
> Recalling SVD can preserve the row and column non-permutability of the original matrix. However, since SVD is an orthogonal unitary equivalence decomposition that requires the row and column orders of the orthogonal matrices to be preserved, it cannot be directly applied in tensor networks.
>
> Consider the low-rank decomposition without orthogonality constraints, i.e., R ≈AC, there exist infinitely many equivalent decomposition forms (A, C)∼ (AD, HC), for DH=I, thus making the identities of the rows impossible to be uniquely determined. Rows and columns are not fixed but drift with D and H with gauge degrees of freedom with $f^2$ parameters.
>
> For R≈ ABC, where B is a diagonal matrix, we have Theorem 1.
>
> Theorem 1.  Without orthogonality constraints, B reduces the gauge degrees of freedom of R ≈AC from the general linear group to the scaling-permutation group with $2f$ parameters.
> Proof. Set ABC = (AH)(JBK)(LC), where HJ=I and KL = I. It can be readily obtained that KBL is a diagonal matrix if and only if both K and L are diagonal matrices. Note that H=$J^{−1}$ and L=$K^{−1}$ , H and L are a diagonal matrices. Therefore, the degrees of freedom for R = ABC are the sum of the scaling degrees of freedom and the permutation degrees of freedom, that is, $2f$.
>
> Introducing B can reduce the gauge degrees of freedom resulting reducing row/column drift, thereby achieving some of the functionalities similar to SVD. It decouples the computation from the parameter tensor while incurring lower computational overhead, making it well-suited for application in tensor networks.
>
> 2.  We have made broader empirical study.
> a) We incorporate two additional state-of-the-art comparative models, namely DGSTA (2025) and Maginet (2025). On D1.2, D2.2, and D3.2, the MAE values of DGSTA are 2.754, 2.454, and 1.722, respectively, while the MAE values of Maginet are 5.067, 2.303, and 0.740, respectively. Compared with ALFT, DGSTA achieves 20.8% lower, 7.8% and 12.8% higher in MAE; meanwhile, Maginet achieves 31.4% and 1.8% higher and 50.7% lower in MAE. Additional models exhibit performance discrepancies on datasets of different scales, and they each possess their own advantages compared with the proposed ALFT, i.e., by leveraging a more powerful dynamic graph convolution module, DGSTA performs better in scenarios with an extremely small number of nodes and Maginet, who works with attention mechanism, performs better in scenarios with an specially large number of nodes.
> b) Beyond traffic datasets, experiments on the movie rating dataset MovieLens , a self-collected sparse video dataset, and the knowledge graphs UMLS and Kinship are conducted. They can be constructed as sparse 3rd-order tensors. In MovieLens, one mode corresponds to an entity graph; in the video dataset, all three modes are entity graphs; whereas in the knowledge graph datasets, none of the modes are entity graphs. The experimental results show that the model with no diagonal matrices activated performs best on the MovieLens and knowledge graphs, while the model with all diagonal matrices activated achieves the best performance on the video dataset. This demonstrates that ANLF can adapt the data based on the judgment of the entity types of the modes.
>
> 3. We will supplement the detailed proofs and experimental results generated during the rebuttal process on the open platform.

---

> > ### Author Rebuttal · Reviewer_kN7Q · 2026-04-03
> >
> > Thank you for your detailed rebuttal and the additional results. However, the theoretical proof provided (Theorem 1) contains fundamental mathematical errors, and the new empirical results contradict the paper’s core motivation. My major concerns remain unresolved.
> >
> > The claim that introducing a diagonal matrix $B$ into an unconstrained factorization $R \approx ABC$ reduces the gauge degrees of freedom from $f^2$ to $2f$ is mathematically incorrect:
> >
> > *   **Gauge freedom remains $O(f^2)$:** Without orthogonality constraints on $A$ and $C$, introducing $B$ does not restrict the general linear group transformation. For any arbitrary invertible matrix $M \in GL(f)$, we can trivially write $R = ABC = (A M^{-1}) I (MBC) = A' I C'$. The system still suffers from the full $f^2$ degrees of freedom; $B$ cannot prevent "row/column drift".
> > *   **False "if and only if" condition:** Your proof states that $JBK$ is diagonal if and only if $J$ and $K$ are diagonal. This is false. If $J$ is a permutation matrix $P$ and $K = P^T$, $PBP^T$ remains diagonal, yet $P$ and $P^T$ are not diagonal matrices.
> > *   **Invalid dimension counting:** The permutation group is a discrete finite group. Adding continuous scaling parameters ($f$) to a discrete permutation group to claim a total of "$2f$ parameters" is mathematically meaningless.

---

> > > ### Author Response · Authors · 2026-04-03
> > >
> > > Thank you for your comments. We apologize for not clarifying your concerns adequately in our previous submission.
> > >
> > > Regarding the theoretical proof:
> > >
> > > 1) Your mathematical derivation is correct in terms of strict matrix operations. However, please note that the determination of $A$, $B$, and $C$ is an optimization process. We impose structural constraints on $ABC$, during optimization, $A$ and $C$ are updated independently and asynchronously ($A$ and $C$ are abstracted from the entity tensor and are also influenced by relations from other modes), while the update of $B$ lags behind $A$ and $C$. Therefore, although the identity $ABC=A'IC′=(AX^{−1} )I(XBC)$ holds mathematically, it is not rigorous in the optimization context, as it treats $BC$ as a single block and is essentially equivalent to $R=AC$. In our proof, we decompose $ABC=(AH)(JBK)(LC)$, which can be interpreted as follows: given updates to $A$ and $C$ via transformations $H$ and $L$, there must exist a unique pair of operations $J$ and $K$ that update B accordingly.
> > >
> > > 2) Your counterexample is mathematically rigorous; however, in ALFT, the conditions that $J$ and $K$ are permutation matrices or  $K=J^T$ are extremely difficult to satisfy.
> > >
> > >
> > > 3) We appreciate your pointing out this issue and will revise our statement as follows:
> > >
> > > $R=ABC$ can reduce the gauge group from the  $f^2$-dimensional continuous group in $R=AC$ to an $f$-dimensional continuous group plus a finite discrete group, thereby significantly decreasing the degrees of freedom, although full discretization as in SVD is not achieved.
> > >
> > > Regarding the additonal experiments:
> > >
> > > Please note that one of the key contributions of this paper is identifying that traditional methods overlook the distinction between entity sets and entity graphs in relational data, which carries important prior information. ALFT leverages this prior to a certain extent by constructing discrepancies in degrees of freedom, and thus can be regarded as an important tool for verifying the existence of the target problem in this work.
> > >
> > > To further verify this point and also address the concerns raised by other reviewers regarding the model's generalizability, we conducted tests on other types of datasets, which impose different requirements on the constraints. Note that the objective function for the knowledge graph dataset differs from that in the paper, using cross-entropy, for consistency, we evaluate results using RMSE and MAE. The results are shown below. It can be observed from the results that the diagonal constraint does possess the ability to regularize the entity graph structure. On the video dataset (where all modes correspond to entity graphs), diagonal matrices strengthen the model’s structural constraint ability, so activating all diagonal matrices yields the best performance.In contrast, on the recommendation and knowledge graph datasets (with no or only one entity graph mode), diagonal matrices have a negative impact.
> > >
> > > RMSE
> > > |   | UMLS | Kinship |  Movielens100K | Sparse Videos |
> > > | :---- | :----: | :----: | :----: |  :----: |
> > > | All diagonal matrices deactivated | 0.248 | 0.291|    0.970|  26.576  |
> > > | One diagonal matrix activated | 0.521 | 0.553|  0.977|   26.510   |
> > > | All diagonal matrices activated | 0.365 | 0.379|   0.974 |    25.555    |
> > >
> > > MAE
> > > |   | UMLS | Kinship |  Movielens100K | Sparse Videos |
> > > | :---- | :----: | :----: | :----: |  :----: |
> > > | All diagonal matrices deactivated | 0.168 | 0.228 |   0.758|  18.728  |
> > > | One diagonal matrix activated | 0.432 | 0.482|  0.765 |   18.321   |
> > > | All diagonal matrices activated | 0.278 | 0.335|   0.762 |     17.849   |

---

### Official Review · Reviewer_koKv · 2026-03-11

**Soundness:** 2
**Presentation:** 3
**Significance:** 3
**Originality:** 3
**Overall Recommendation:** 4
**Confidence:** 4

**Summary:**

This paper discusses a method for estimating relationships from relational data represented as a higher-order tensor by learning latent features. Conventional Latent Factorization-of-Tensor (LFT) methods implicitly assume that the entities in each mode form a simple set without internal structure. However, in many real-world scenarios, entities exhibit additional structure such as ordering or graph relationships, which conventional LFT methods cannot adequately capture. To address this issue, the authors distinguish between two types of modes: entity sets, where entities are treated as an unordered collection, and entity graphs, where entities possess internal structural relationships. The proposed method models the interactions between entity graphs as an aligned bipartite network. Specifically, diagonal matrix constraints are imposed between the latent tensors of entity graph modes, which is expected to preserve structural properties in a manner analogous to singular value decomposition (SVD). In addition, the model introduces a mechanism that imposes stronger constraints on entities with a larger amount of data. Experimental evaluations on real-world datasets demonstrate that the proposed method achieves superior performance in tensor reconstruction compared with existing higher-order relational learning methods.

**Compliance With Llm Reviewing Policy:**

Affirmed.

**Final Justification:**

I appreciate the authors’ response, especially the clarification on how the diagonal constraint reduces gauge freedom, which helps address my concern about identifiability.

That said, I still find it unclear how to distinguish between entity sets and entity graphs in practice. The explanation seems to rely on prior knowledge or manual judgment, which may not be straightforward in real-world situations. Since this directly affects when the method can be applied, I think this point should be clearly acknowledged as a limitation.

Overall, I find the problem setting and the idea behind the method interesting. With the above clarification added to the paper, I am willing to revise my score to a weak accept.

**Key Questions For Authors:**

1. Could the authors further discuss how the diagonal matrix constraint helps capture the underlying internal structure? In addition, could the authors clarify whether this constraint enables the model to identify such structures uniquely?

2. Could the authors discuss how to distinguish between an entity set and an entity graph in practice?

**Limitations:**

The proposed method appears to require at least two modes, corresponding to entity graphs, to be applicable. If this interpretation is correct, it could represent a significant limitation of the approach. It would be helpful if the authors could clarify how the method could be applied when only a single mode corresponds to an entity graph. Alternatively, suggesting an extension of the method or providing an alternative approach for such cases could help mitigate this limitation.

**Strengths And Weaknesses:**

Strengths:

- The paper clearly points out that modes in relational tensors may consist of a mixture of entity sets and entity graphs, and highlights that conventional Latent Factorization-of-Tensor (LFT) methods implicitly assume all modes are simple sets. Identifying this limitation and formulating it as a research problem constitutes a meaningful contribution to the community working on higher-order relational learning.

- The paper proposes a simple and intuitive idea: imposing diagonal matrix constraints between latent tensors so that the interaction structure resembles that of singular value decomposition (SVD). This idea is conceptually easy to understand.

- Experimental evaluations on real-world data demonstrate that the proposed method achieves higher performance in tensor reconstruction compared with several existing higher-order relational learning approaches.

Weaknesses:

- While the idea of introducing diagonal constraints between latent factors to obtain an SVD-like effect is interesting, the paper does not provide a concrete discussion of how this constraint enables the model to capture internal structural relationships among entities. As a result, it is unclear under what conditions the proposed method is expected to be effective. A deeper discussion of this point could clarify the applicability of the method and help define the types of data for which it is suitable.

- The paper does not provide a theoretical analysis or guarantees showing that the proposed method can successfully capture the underlying internal structure. Even if the model can recover such structures in practice, it remains unclear whether they can be uniquely identified. Intuitively, if several diagonal elements of the constraint matrix take similar values, the corresponding latent factors may still retain substantial rotational ambiguity. Discussing this issue could significantly improve the credibility and reliability of the proposed approach.

- The paper does not discuss how to distinguish between entity sets and entity graphs in practice. From the description, this distinction appears to rely on manual judgment. However, in real-world applications, this decision may be difficult. For example, in purchase data, it may be unclear whether latent structural relationships among products or customers should be assumed. Providing criteria or indicators to guide this decision could substantially enhance the practical usefulness of the proposed method.

---

> ### Author Rebuttal · Authors · 2026-03-30
>
> Thank you for your comment.
> 1.  To verify the correctness of the theory proposed in this paper, we have made the following efforts.
>
> Proposition 1. In a relational tensor, the entities corresponding to a mode may be either set or labeled entities. In R, the former manifests as rows and columns that can be permuted arbitrarily, whereas the latter have fixed rows and columns.
>
> Proposition 1 is self-evident, for example shuffling the temporal order of time-varying data or altering the rows and columns of an image corrupts the original data.
>
> Proving the theoretical correctness of the proposed method is equivalent to demonstrating that introducing a learnable diagonal matrix can enhance the model's ability to learn and preserve the structural invariance of the mode pair R. To simplify the proof, we assume that the parameter tensor is the contraction of a series of matrices, among which one matrix can characterize the relationship between the current mode and the other mode associated with R, and the relationship R can be approximated by a series of matrix multiplications.
>
> Recalling SVD can preserve the row and column non-permutability of the original matrix. However, since SVD is an orthogonal unitary equivalence decomposition that requires the row and column orders of the orthogonal matrices to be preserved, it cannot be directly applied in tensor networks.
>
> Consider the low-rank decomposition without orthogonality constraints, i.e., R ≈AC, there exist infinitely many equivalent decomposition forms (A, C)∼ (AD, HC), for DH=I, thus making the identities of the rows impossible to be uniquely determined. Rows and columns are not fixed but drift with D and H with gauge degrees of freedom with $f^2$ parameters.
>
> For R≈ ABC, where B is a diagonal matrix, we have Theorem 1.
>
> Theorem 1.  Without orthogonality constraints, B reduces the gauge degrees of freedom of R ≈AC from the general linear group to the scaling-permutation group with $2f$ parameters.
> Proof. Set ABC = (AH)(JBK)(LC), where HJ=I and KL = I. It can be readily obtained that KBL is a diagonal matrix if and only if both K and L are diagonal matrices. Note that H=$J^{−1}$ and L=$K^{−1}$ , H and L are a diagonal matrices. Therefore, the degrees of freedom for R = ABC are the sum of the scaling degrees of freedom and the permutation degrees of freedom, that is, $2f$.
>
> Introducing B can reduce the gauge degrees of freedom resulting reducing row/column drift, thereby achieving some of the functionalities similar to SVD. It decouples the computation from the parameter tensor while incurring lower computational overhead, making it well-suited for application in tensor networks.
>
> 2.  It should be clarified that due to structural constraints of ANLF, the diagonal matrix cannot impose a uniqueness constraint; it only serves to reduce the degrees of freedom for errors on the premise of not interfering with other mode pairs. In practice, an entity graph can be determined by checking whether the rows/columns of the adjacency matrix corresponding to R are interchangeable.
>
> 3.  When only one mode corresponds to the entity graph, ANLF can also function effectively. The activation conditions of the diagonal matrix between this mode and other modes can be derived as follows.
>
> The relationship R between two modes is represented as a bipartite graph G=(U,I,E), where U is an entity graph and I is an entity set.  For approximation $R \approx \sum_{k=1}^{f} b_{kk} \cdot \mathbf{a}_k \mathbf{c}_k^T$, let $w_i$ denote the local importance vector of entity $u_i$.
>
> When B is not activated, $a_i=w_i \odot t_i$, where $t_i$ denotes the basis representation of entity i, and $\odot$ stands for the element-wise product. That is, $w_i$ is required to be hard-coded into each row of A.
>
> When B is activated, $a_i= \sum_{k=1}^{f} b_{kk} \cdot t_{ik} \cdot c_k$, where $b_{kk}$ provides global regulation. Let $E_U$, $\hat{w}$, and $E_W$ be the row covariance of A, average local importance and variance of importance, respectively. The gain from adding B is proportional to $\|E_W\|_F \cdot \|E_U\|_F$.
>
> Define the heterogeneity index $\eta := \frac{\|E_W\|_F}{\|\hat{w}\|^2}$. If $\eta$ exceeds a certain threshold, adding B is beneficial; otherwise, B is redundant and we can directly adopt R=AC.

---

> > ### Author Rebuttal · Reviewer_koKv · 2026-04-02
> >
> > I appreciate the authors’ response, especially the clarification on how the diagonal constraint reduces gauge freedom, which helps address my concern about identifiability.
> >
> > That said, I still find it unclear how to distinguish between entity sets and entity graphs in practice. The explanation seems to rely on prior knowledge or manual judgment, which may not be straightforward in real-world situations. Since this directly affects when the method can be applied, I think this point should be clearly acknowledged as a limitation.
> >
> > Overall, I find the problem setting and the idea behind the method interesting. With the above clarification added to the paper, I am willing to revise my score to a weak accept.

---

> > > ### Author Response · Authors · 2026-04-03
> > >
> > > Thank you very much for your comment. As you pointed out, although we have a criterion for determining whether a mode corresponds to an entity set or an entity graph, we currently rely on manual judgment to adjust the embedding structure of ALFT. We will include this point in the limitations and also plan to explore adaptive methods in future work.

---

### Official Review · Reviewer_Zfmv · 2026-03-13

**Soundness:** 3
**Presentation:** 3
**Significance:** 3
**Originality:** 3
**Overall Recommendation:** 5
**Confidence:** 2

**Summary:**

**Disclaimer:** I am not an expert in tensor factorization and tensor network, so this is only a review based on my current understanding of the paper.

The paper works on relation extraction from relational tensors. The motivation is that existing Latent Factorization-of-Tensors methods usually assume each mode is an unordered entity set and that relations between mode pairs are simple bipartite graphs. But this asummption is not suitable when some modes have internal topology, such as time or coordinates, where the relation between two modes should be viewed as a more structured aligned bipartite network.

To address this, the paper proposes Asymmetric Latent Factorization-of-Tensors (ALFT). The key idea is to start from a fully-connected tensor network style factorization and insert learnable diagonal matrices between selected mode pairs, so that topology-sensitive pairs can be constrained in an SVD-like way while ordinary set-like mode pairs remain unconstrained.

**Compliance With Llm Reviewing Policy:**

Affirmed.

**Key Questions For Authors:**

1. How well does the approach generalize beyond traffic datasets to other relational tensors with nontrivial topology?

2. Can the paper compare with stronger modern neural or graph-based baselines designed for spatiotemporal modeling?

**Limitations:**

1. The broader usefulness beyond this specific application domain is not fully demonstrated.

**Strengths And Weaknesses:**

## Soundness:

**Strentgh:**
1. The paper has a clear motivation: not all tensor modes should be treated as plain unordered sets, and this is a reasonable point especially for temporal or spatial modes.

2. The method is technically simple and coherent. The proposed model can be understood as FCTN plus diagonal pairwise constraints, and the optimization objective is straightforward.

3. The experiments include ablations on different constraint placements, which is helpful for supporting the claims.

## Presentation:

**Strentgh:**
1. The paper is generally readable and the high-level intuition can be followed.

## Significance:

**Strentgh:**
1. The paper studies a potentially meaningful issue in tensor relational modeling: some modes may have internal structure and should perhaps not be modeled identically to ordinary entity sets.

2. The approach can be useful for spatiotemporal tensor completion tasks.

**Weakness:**
1. The practical impact seems limited by the evaluation scope. The experiments are mainly on several traffic datasets, which makes it hard to know how broadly the idea generalizes.

2. The paper does not compare against stronger modern structured models (like Graph Neural Network) that one may expect for spatiotemporal data.


## Originality:

**Strentgh:**
1. The paper gives a specific perspective that different mode pairs may deserve different latent interaction structures.

2.The asymmetric constraint design is a reasonable extension of FCTN.

**Weakness:**
1. The overall originality looks moderate. The work mainly combines existing tensor-network factorization ideas with SVD-inspired diagonal constraints.

---

> ### Author Rebuttal · Authors · 2026-03-30
>
> Thank you for your comment.
> 1. We have made broader empirical study.
> a) We incorporate two additional state-of-the-art comparative models, namely DGSTA (2025) and Maginet (2025). On D1.2, D2.2, and D3.2, the MAE values of DGSTA are 2.754, 2.454, and 1.722, respectively, while the MAE values of Maginet are 5.067, 2.303, and 0.740, respectively. Compared with ALFT, DGSTA achieves 20.8% lower, 7.8% and 12.8% higher in MAE; meanwhile, Maginet achieves 31.4% and 1.8% higher and 50.7% lower in MAE. Additional models exhibit performance discrepancies on datasets of different scales, and they each possess their own advantages compared with the proposed ALFT, i.e., by leveraging a more powerful dynamic graph convolution module, DGSTA performs better in scenarios with an extremely small number of nodes and Maginet, who works with attention mechanism, performs better in scenarios with an specially large number of nodes.
> b) Beyond traffic datasets, experiments on the movie rating dataset MovieLens , a self-collected sparse video dataset, and the knowledge graphs UMLS and Kinship are conducted. They can be constructed as sparse 3rd-order tensors. In MovieLens, one mode corresponds to an entity graph; in the video dataset, all three modes are entity graphs; whereas in the knowledge graph datasets, none of the modes are entity graphs. The experimental results show that the model with no diagonal matrices activated performs best on the MovieLens and knowledge graphs, while the model with all diagonal matrices activated achieves the best performance on the video dataset. This demonstrates that ANLF can adapt the data based on the judgment of the entity types of the modes.
>
> 2. The findings of this work can introduce new priors for exploring deep connections in entity interactions.  This structure can be effectively applied to the analysis of high-dimensional interaction relationships, while also providing a meaningful constraint for tensor structure search. We also verify the correctness of the theory proposed in this paper to make it more solid.
>
> Proposition 1. In a relational tensor, the entities corresponding to a mode may be either set or labeled entities. In R, the former manifests as rows and columns that can be permuted arbitrarily, whereas the latter have fixed rows and columns.
>
> Proposition 1 is self-evident, for example shuffling the temporal order of time-varying data or altering the rows and columns of an image corrupts the original data.
>
> Proving the theoretical correctness of the proposed method is equivalent to demonstrating that introducing a learnable diagonal matrix can enhance the model's ability to learn and preserve the structural invariance of the mode pair R. To simplify the proof, we assume that the parameter tensor is the contraction of a series of matrices, among which one matrix can characterize the relationship between the current mode and the other mode associated with R, and the relationship R can be approximated by a series of matrix multiplications.
>
> Recalling SVD can preserve the row and column non-permutability of the original matrix. However, since SVD is an orthogonal unitary equivalence decomposition that requires the row and column orders of the orthogonal matrices to be preserved, it cannot be directly applied in tensor networks.
>
> Consider the low-rank decomposition without orthogonality constraints, i.e., R ≈AC, there exist infinitely many equivalent decomposition forms (A, C)∼ (AD, HC), for DH=I, thus making the identities of the rows impossible to be uniquely determined. Rows and columns are not fixed but drift with D and H with gauge degrees of freedom with $f^2$ parameters.
>
> For R≈ ABC, where B is a diagonal matrix, we have Theorem 1.
>
> Theorem 1.  Without orthogonality constraints, B reduces the gauge degrees of freedom of R ≈AC from the general linear group to the scaling-permutation group with $2f$ parameters.
> Proof. Set ABC = (AH)(JBK)(LC), where HJ=I and KL = I. It can be readily obtained that KBL is a diagonal matrix if and only if both K and L are diagonal matrices. Note that H=$J^{−1}$ and L=$K^{−1}$ , H and L are a diagonal matrices. Therefore, the degrees of freedom for R = ABC are the sum of the scaling degrees of freedom and the permutation degrees of freedom, that is, $2f$.
>
> Introducing B can reduce the gauge degrees of freedom resulting reducing row/column drift, thereby achieving some of the functionalities similar to SVD. It decouples the computation from the parameter tensor while incurring lower computational overhead, making it well-suited for application in tensor networks.

---

> > ### Author Rebuttal · Reviewer_Zfmv · 2026-04-01
> >
> > Thanks for your response, but it seems that the results of new experiments is not shown. I am still not confident with current judgement, so i decide to keep the result.

---

> > > ### Author Response · Authors · 2026-04-03
> > >
> > > Thank you for your valuable feedback and encouragement. Taking this final opportunity to communicate with you, we would like to reiterate the focus of our work.
> > >
> > > The core effort of this paper is to identify an overlooked issue, i.e., the distinction between entity sets and entity graphs in relational data, which we argue serves as effective prior information.
> > >
> > > Building upon existing tensor network methods, we design a model and prove that it can regulate the injection of this prior by reducing the degrees of freedom between specific mode pairs.
> > >
> > > To further verify the existence of the target issue, beyond the transportation dataset, we have conducted empirical analyses on various datasets with different constraint scenarios. The additional experimental results on these new datasets are presented below:
> > >
> > > RMSE:
> > > |   | UMLS | Kinship |  Movielens100K | Sparse Videos |
> > > | :---- | :----: | :----: | :----: |  :----: |
> > > | All diagonal matrices deactivated | 0.248 | 0.291|    0.970|  26.576  |
> > > | One diagonal matrix activated | 0.521 | 0.553|  0.977|   26.510   |
> > > | All diagonal matrices activated | 0.365 | 0.379|   0.974 |    25.555    |
> > >
> > > MAE:
> > > |   | UMLS | Kinship |  Movielens100K | Sparse Videos |
> > > | :---- | :----: | :----: | :----: |  :----: |
> > > | All diagonal matrices deactivated | 0.168 | 0.228 |   0.758|  18.728  |
> > > | One diagonal matrix activated | 0.432 | 0.482|  0.765 |   18.321   |
> > > | All diagonal matrices activated | 0.278 | 0.335|   0.762 |     17.849   |
> > >
> > > There datasets impose different requirements on the constraints. Note that the objective function for the knowledge graph dataset differs from that in the paper, using cross-entropy, for consistency, we evaluate results using RMSE and MAE. The results are shown below. It can be observed from the results that the diagonal constraint does possess the ability to regularize the entity graph structure. On the video dataset (where all modes correspond to entity graphs), diagonal matrices strengthen the model's structural constraint ability, so activating all diagonal matrices yields the best performance.In contrast, on the recommendation and knowledge graph datasets (with no or only one entity graph mode), diagonal matrices have a negative impact. These results complement our findings on the transportation dataset, where two modes correspond to entity graphs, and activating one diagonal matrix achieved the best performance.

---

### Official Review · Reviewer_fN2V · 2026-03-13

**Soundness:** 2
**Presentation:** 3
**Significance:** 3
**Originality:** 3
**Overall Recommendation:** 4
**Confidence:** 2

**Summary:**

This paper identifies a novel limitation in existing Latent Factorization-of-Tensors (LFT) models for relation extraction, specifically their inability to handle aligned bipartite networks where entities within a mode have ordered topological structures. The authors propose an Asymmetric LFT (ALFT) model to address this gap by imposing constraints between specific mode pairs. Experimental results validate the issue's existence and demonstrate the effectiveness of the proposed solution.

**Compliance With Llm Reviewing Policy:**

Affirmed.

**Final Justification:**

The authors' responses during this rebuttal effectively resolved some of my concerns. I fully appreciate the efforts made by the authors and the AC, and I hope my comments can serve as a reference for them.

**Key Questions For Authors:**

Please see the weaknesses.

**Limitations:**

yes

**Strengths And Weaknesses:**

strengths
The manuscript points out that the existing Latent Factorization-of-Tensors models usually assume that each mode of the tensor corresponds to an unstructured entity set, thereby ignoring the inherent topological structure of entities such as time series or spatial coordinates. To solve the Graph Isomorphism problem caused by the misuse of bipartite graph structure, the authors propose an Asymmetric Tensor Network, which introduces a learnable diagonal constraint matrix between specific mode pairs to maintain structural stability.

weaknesses:
1. As stated in the overview, the experimental subjects of this paper are traffic flow data. In the field of machine learning, tensor decomposition is widely used for estimating and completing the missing values of such multi-dimensional data. However, the author regards it as Relation Extraction. In natural language processing, relation extraction aims to convert unstructured text into a triplet structure. These two are completely different in terms of data manifold, noise characteristics, and evaluation metrics. This packaging not only fails to highlight the advantages of tensor networks in temporal and spatial modeling, but also tends to raise doubts about the author and lack the most basic understanding of the classic entity relation extraction paradigm in the NLP field. I think the author needs to clarify this issue.
2. In Section 3.1, the manuscript proposes to insert a learnable diagonal matrix $M_{p,q}$ between the potential feature tensors of modalities $p$ and $q$, and claims that this "SVD-like structure" can break the graph isomorphism trap caused by the entity set assumption and maintain the interaction topology. This is difficult to form an evidence chain in mathematical logic. The diagonal matrix geometrically represents only linear scaling along each coordinate axis, and it does not involve rotation or Permutation. If the original tensor factorization has structural isomorphism invariance, how can the simple feature dimension scaling be strictly proved to break this invariance mathematically? Therefore, the authors must provide rigorous mathematical theorems and Mathematical Proofs to prove that after introducing $M_{p,q}$ into the tensor network, the new operator strictly bounds the Expressiveness Capacity when capturing the Ordered Time-slice Graphs structure, rather than merely relying on intuitive textual descriptions.
3. Table 2 lists the comparison models such as NCF (2017) and STD (2018), which have become seriously outdated. As for the field of tensor completion, we should compare the recent state-of-the-art prediction models based on spatio-temporal graph convolution networks (ST-GCN) or spatio-temporal attention mechanisms.

---

> ### Author Rebuttal · Authors · 2026-03-30
>
> 1. Thank you very much for pointing out the terminological issue. We intend to express is the extraction of interaction information between arbitrary entities, which constitutes a generalized form of relation extraction, and relation extraction in NLP is merely one specific scenario of it. We will explicitly clarify this point in the revised version of the paper.
>
> 2. Thank you for your comment.  To verify the correctness of the theory proposed in this paper, we have made the following efforts.
>
> Proposition 1. In a relational tensor, the entities corresponding to a mode may be either set or labeled entities. In R, the former manifests as rows and columns that can be permuted arbitrarily, whereas the latter have fixed rows and columns.
>
> Proposition 1 is self-evident, for example shuffling the temporal order of time-varying data or altering the rows and columns of an image corrupts the original data.
>
> Proving the theoretical correctness of the proposed method is equivalent to demonstrating that introducing a learnable diagonal matrix can enhance the model's ability to learn and preserve the structural invariance of the mode pair R. To simplify the proof, we assume that the parameter tensor is the contraction of a series of matrices, among which one matrix can characterize the relationship between the current mode and the other mode associated with R, and the relationship R can be approximated by a series of matrix multiplications.
>
> Recalling SVD can preserve the row and column non-permutability of the original matrix. However, since SVD is an orthogonal unitary equivalence decomposition that requires the row and column orders of the orthogonal matrices to be preserved, it cannot be directly applied in tensor networks.
>
> Consider the low-rank decomposition without orthogonality constraints, i.e., R ≈AC, there exist infinitely many equivalent decomposition forms (A, C)∼ (AD, HC), for DH=I, thus making the identities of the rows impossible to be uniquely determined. Rows and columns are not fixed but drift with D and H with gauge degrees of freedom with $f^2$ parameters.
>
> For R≈ ABC, where B is a diagonal matrix, we have Theorem 1.
>
> Theorem 1.  Without orthogonality constraints, B reduces the gauge degrees of freedom of R ≈AC from the general linear group to the scaling-permutation group with $2f$ parameters.
> Proof. Set ABC = (AH)(JBK)(LC), where HJ=I and KL = I. It can be readily obtained that KBL is a diagonal matrix if and only if both K and L are diagonal matrices. Note that H=$J^{−1}$ and L=$K^{−1}$ , H and L are a diagonal matrices. Therefore, the degrees of freedom for R = ABC are the sum of the scaling degrees of freedom and the permutation degrees of freedom, that is, $2f$.
>
> Introducing B can reduce the gauge degrees of freedom resulting reducing row/column drift, thereby achieving some of the functionalities similar to SVD. It decouples the computation from the parameter tensor while incurring lower computational overhead, making it well-suited for application in tensor networks.
>
> 3.  We have made broader empirical study.
> a) We incorporate two additional state-of-the-art comparative models, namely DGSTA (2025) and Maginet (2025). On D1.2, D2.2, and D3.2, the MAE values of DGSTA are 2.754, 2.454, and 1.722, respectively, while the MAE values of Maginet are 5.067, 2.303, and 0.740, respectively. Compared with ALFT, DGSTA achieves 20.8% lower, 7.8% and 12.8% higher in MAE; meanwhile, Maginet achieves 31.4% and 1.8% higher and 50.7% lower in MAE. Additional models exhibit performance discrepancies on datasets of different scales, and they each possess their own advantages compared with the proposed ALFT, i.e., by leveraging a more powerful dynamic graph convolution module, DGSTA performs better in scenarios with an extremely small number of nodes and Maginet, who works with attention mechanism, performs better in scenarios with an specially large number of nodes.
> b) Beyond traffic datasets, experiments on the movie rating dataset MovieLens , a self-collected sparse video dataset, and the knowledge graphs UMLS and Kinship are conducted. They can be constructed as sparse 3rd-order tensors. In MovieLens, one mode corresponds to an entity graph; in the video dataset, all three modes are entity graphs; whereas in the knowledge graph datasets, none of the modes are entity graphs. The experimental results show that the model with no diagonal matrices activated performs best on the MovieLens and knowledge graphs, while the model with all diagonal matrices activated achieves the best performance on the video dataset. This demonstrates that ANLF can adapt the data based on the judgment of the entity types of the modes.

---

> > ### Author Rebuttal · Reviewer_fN2V · 2026-04-02
> >
> > I thank the authors for their nice response.
> >
> > However, some of my concerns remain unresolved. In tensor networks, gauged degrees of freedom allow the insertion of any invertible matrix $X$ and its inverse $X^{-1}$ into the bond dimension connecting two tensors without altering the global tensor state. The authors claim that introducing $B$ can reduce row/column drift, achieving functionality similar to SVD. However, the core value of SVD lies in the orthogonality constraints it provides, guaranteeing the uniqueness of basis vectors and the optimal distribution of energy. The authors' "Theorem 1" only solves the scaling problem but fails to address the degrees of freedom introduced by rotation. Without orthogonality constraints, even if the degrees of freedom decrease from $f^2$ to $2f$, a large non-unique space still exists between tensor cores. Modern tensor ring decomposition studies (such as BLOSTR) show that to achieve accurate core recovery, blockwise simultaneous diagonalization must be utilized instead of simple diagonal matrix insertion. Furthermore, any insertion of diagonal matrices between factor matrices formally reduces the number of parameters, but this does not equate to enhancing the model's ability to learn "structural invariance." True structural invariance requires fixing the coordinate system through an orthogonalization process, similar to singular value decomposition, a mechanism that the ALFT framework appears to lack.
> >
> > If the author resolves this question for me, I will raise my score.

---

> > > ### Author Response · Authors · 2026-04-03
> > >
> > > We appreciate your insightful question.
> > > To clarify, as you pointed out and as stated in Theorem 1, ALFT cannot fully fix the entity graph structure without orthogonal constraints. This is because ALFT uses tensor networks as its decomposition backbone. In a tensor network, the parameter tensor $T_ i$ corresponding to mode $i$ interacts with all other mode-specific parameter tensors. Although imposing orthogonal constraints can regulate the relational structure between mode $i$ and specific modes, full SVD-style orthogonalization would couple the singular vector matrices of different mode pairs within $T_i$, thereby introducing unintended indirect constraints on mode pairs that do not require such restrictions. Therefore, the introduction of the diagonal matrix essentially strikes a balance between gauge degrees of freedom and representation ability. By constructing differences in degrees of freedom across different types of mode pairs, it incorporates entity-type priors in a manner analogous to regulating osmotic pressure, i.e., it is a relative structural constraint rather than an absolute structural constraint.
> > >
> > > Please note that one of the key contributions of this paper is identifying that traditional methods overlook the distinction between entity sets and entity graphs in relational data, which carries important prior information. ALFT leverages this prior to a certain extent by constructing discrepancies in degrees of freedom, and thus can be regarded as an important tool for verifying the existence of the target problem in this work. In the additional experiments, we employed different structures to reconstruct video data (32 frames) with 60% of pixels randomly removed. The experimental results show that the model achieves the best performance when all diagonal matrices are activated, which indirectly demonstrates that the diagonal matrices have a certain ability to preserve entity structures.
> > >
> > > |   | Average RMSE | Average MAE |
> > > | :---- | :----: | :----: |
> > > | All diagonal matrices deactivated |  26.576  | 18.728|
> > > | One diagonal matrix activated |   26.510   | 18.321|
> > > | All diagonal matrices activated |    25.555    |  17.849  |
> > >
> > > Given the structure of tensor networks, we believe it is difficult to directly impose orthogonal constraints. We have carefully read the work A Provably Efficient Method for Tensor Ring Decomposition and Its Applications. The blockwise simultaneous diagonalization method is highly elegant, yet it cannot be directly applied to general tensor networks at present, since the information coupled inside the parameter tensors after tensor network decomposition is more complex than that in tensor ring decomposition. This ingenious approach has inspired us and points out a direction for our future work.
> > >
> > > We thank you again for this highly insightful comment, which has helped us re-examine the objectives and scope of our work. We will clarify the limitations of the current model and our future goals in the revised paper.

---

### Decision · Program_Chairs · 2026-04-30

**Decision:**

Accept (regular)

**Comment:**

The reviewers agreed that the paper identifies an interesting and meaningful limitation of existing LFT methods by distinguishing between entity sets and entity graphs, and found the proposed asymmetric diagonal-constraint design intuitive and well-motivated. The additional experiments provided in the rebuttal further strengthened the empirical evidence, suggesting that the method can adapt to different structural settings and offering useful practical insights. At the same time, some concerns remain regarding the theoretical interpretation, particularly whether the diagonal constraint achieves SVD-like identifiability or should be viewed as a weaker structural prior; the rebuttal clarified this point but did not fully resolve all questions. Reviewers also noted that the distinction between entity sets and entity graphs relies on prior knowledge and should be clearly stated as a limitation. Overall, the paper is considered to present a promising and practically relevant idea, with improved empirical support after rebuttal, though with some remaining questions on theoretical rigor and scope.